# The Design of CNN Architectures for Optimal Six Basic Emotion Classification Using Multiple Physiological Signals

**DOI:** 10.3390/s20030866

**Published:** 2020-02-06

**Authors:** SeungJun Oh, Jun-Young Lee, Dong Keun Kim

**Affiliations:** 1Department of Sports ICT Convergence, Sangmyung University Graduate School, Seoul 03016, Korea; seungjun0215@gmail.com; 2Department of Psychiatry and Neuroscience Research Institute, Seoul National University College of Medicine, SMG-SNU Boramae Medical Center, Seoul 07061, Korea; benji@snu.ac.kr; 3Department of Intelligent Engineering Informatics for Human, Institute of Intelligent Informatics Technology, Sangmyung University, Seoul 03016, Korea

**Keywords:** emotion classification, physiological signals, machine learning, deep learning, principal components analysis, convolution neural networks

## Abstract

This study aimed to design an optimal emotion recognition method using multiple physiological signal parameters acquired by bio-signal sensors for improving the accuracy of classifying individual emotional responses. Multiple physiological signals such as respiration (RSP) and heart rate variability (HRV) were acquired in an experiment from 53 participants when six basic emotion states were induced. Two RSP parameters were acquired from a chest-band respiration sensor, and five HRV parameters were acquired from a finger-clip blood volume pulse (BVP) sensor. A newly designed deep-learning model based on a convolutional neural network (CNN) was adopted for detecting the identification accuracy of individual emotions. Additionally, the signal combination of the acquired parameters was proposed to obtain high classification accuracy. Furthermore, a dominant factor influencing the accuracy was found by comparing the relativeness of the parameters, providing a basis for supporting the results of emotion classification. The users of this proposed model will soon be able to improve the emotion recognition model further based on CNN using multimodal physiological signals and their sensors.

## 1. Introduction

The negative stimuli that people experience in their daily lives vary widely, and the accumulation of these negative stimuli causes stress and mental illnesses such as depression. However, owing to the lack of awareness of the dangers of mental illness and the negative social perception, treatment behaviors are often passive. Additionally, efforts to alleviate negative stimuli are ongoing, but it is difficult to grasp the degree of stress of an individual, and, in many cases, emotional differences make assessment difficult. Therefore, differences in personal emotions are very important in relieving stress and treating mental illnesses.

Representative techniques for recognizing emotions include image-based recognition, voice-based recognition, and physiological-signal-based recognition [1]. All three techniques show good emotion-recognition performance. However, image-based recognition is influenced by the background or brightness of the image, and recognition is impossible with images taken at some viewing angles. Voice-based emotion recognition identifies instantaneous emotion with high accuracy. However, it has the disadvantage of recognizing emotions only over short time periods [2]. Emotion recognition based on physiological signals is less influenced by the external environment than image- and voice-based recognition. Further, it is less sensitive to sociocultural differences among users. In addition, because physiological signals are controlled by the autonomic nervous system, they cannot be intentionally manipulated. The use of physiological signals is advantageous in that these signals are acquired in a natural emotional state that is learned socially, rather than artificially [3,4]. In other words, emotion recognition using physiological signals is a very powerful method for grasping human emotional states. Moreover, numerous studies have shown that physiological signals and human emotional states are closely related [5,6,7,8]. Based on this reference, the study was conducted using heart rate variability and respiration signal, representative responses of the autonomic nervous system.

Emotion classification through defined emotions and measured signals is performed in various ways, ranging from traditional statistical methods and machine learning (linear regression, logistic regression, hidden Markov model, naïve Bayes classification, support vector machine, and Decision Tree) to the latest technique, deep learning. In particular, the support vector machine (SVM) is a widely used method for emotion classification [9,10,11]. Torres-Valencia et al. [12] classified two-dimensional emotions by using HMM, and studies using the C4.5 Decision Tree [13], *K*-nearest neighbor (KNN) [14,15], and Linear Discriminant Analysis (LDA) [16] have been reported. There have also been studies using deep-learning methods, such as convolutional neural networks (CNNs) [17,18], Deep Belief Network (DBN) [19], and Sparse AE [20], or models integrating machine learning and deep learning [21,22]. In addition to the simple categorization of emotions, studies that apply classified models to various fields have been actively conducted [23,24,25,26]. Even the conventional classifiers used in the past are not inferior in performance to state-of-the-art classifiers, and the results may vary depending on the classifier parameters and the characteristics of input data. Although many classification models have been introduced, this study used the CNN model to classify emotions. CNN is suitable model for processing and recognizing images. Thus, we used the advantages of CNN and applied the structure to classify physiological signals. CNN is a deep learning method and it does not need feature extraction because it learns features automatically. CNN is more scalable. In particular, CNN is easy to integrate with other deep learning models. By transforming the learning method in parallel, it can easily build hybrid or ensembled models with other deep learning models. it can also re-train CNNs for new tasks based on existing models.

As mentioned above, previous studies classified emotions by acquiring physiological signals and then classifying these signals in various ways. In other words, the research focus was on the processing of data and classification of signals using classification models. In contrast, the present study proposes an optimum emotion classification model beyond simple emotion classification. For this purpose, we compared the classification accuracy of single-signal and multi-signal approaches, experimented with various combinations between input signals for classify emotions into various stages, and verified the dominance of parameters using principal component analysis. Furthermore, the emotion recognition methods using various deep-learning techniques proposed in this study are expected to be referred to classify emotions efficiently. The overall flow of this study is shown in Figure 1.

## 2. Related Works

### 2.1. Definition of Emotion and Study Using Physiological Signal

In many studies related to existing emotions, human emotions are divided into arousal and valence [27] by using a two-dimensional emotion classification model based on the circumplex model of affect [28], and dominance is applied to two- and three-dimensionally extended emotional models. This study, however, was based on Ekman’s six basic emotions, which are defined based on facial expressions that are commonly found in various countries and cultures and are associated with happiness, fear, surprise, anger, sadness, and disgust [29].

Emotions defined in this way can be measured using physiological signals. Physiological signals can be easily measured using non-invasive sensors and wearable devices, and various signals such as electrocardiogram (ECG), electromyogram (EMG), electroencephalogram (EEG), galvanic skin response (GSR), skin temperature (SKT) and respiration (RSP) are closely related to human emotions [20]. Different studies used different numbers of signals; Garcia et al. [30] used seven physiological signals to classify emotions, while Zheng et al. [31] and Rami et al [32] classified emotions with EEG signals alone. Many studies have been conducted to classify emotions by combining various types of signals or by obtaining two or more signals, the multimodal signal. Basically, multimodal signals include the use of signal components with plural sensory modalities [33]. There are various ways to apply the multimodal signal. Emotion recognition and analysis were performed using facial expressions and EEG signals by Huang et al. [34] and using facial expressions and voices by Andrea et al. [35]. There is no relationship between the number of signal types and the classification rate, and even when two experiments are performed under the same conditions, the results may vary depending on the selection of features in the analysis.

### 2.2. Emotion Analysis Method

To analyze emotions through respiration, the subject’s respiration should be measured in real time. There are many ways to measure respiration, including the direct measurement of air flow during inhalation and exhalation, the use of a photoplethysmogram (PPG) sensor, and a method of measuring changes in the width of the thoracic cavity by using a respiratory band. The direct measurement of air flow allows the precise measurement of breathing volume, but it involves specific testing equipment and requires subjects to wear inconvenient airflow transducers. As an alternative to this problem, PPG sensors, which use peripheral circulation changes through respiration, may be used [36,37]. However, since the PPG sensor was originally developed for measuring heartbeat, it is necessary to extract the respiratory signal from the PPG signal. However, data loss or distortion may occur in the signal-extraction process, which may be somewhat less accurate than direct measurement of respiration. Therefore, in this study, we decided to measure the size change of the thoracic cavity by utilizing a wearable breathing band to compensate for the limit of secondary data that can be extracted from a PPG sensor. The respiration band measures the change in a magnetic-field sensor built into the band, and it converts the change in size of the chest cavity for each breath into an electrical signal to measure respiration. Unlike the PPG sensor, the measurement of respiration using respiration bands does not require additional signal processing and is simpler than airflow measurement. Therefore, in this study, respiration signals were measured using a respiration band, the respiration rate (RSP rate) was measured from the respiration signal (RSP value), and the measured variables were used as RSP parameters.

Emotions can also be recognized using heartbeat signals in addition to respiratory signals. Heartbeat signals can be measured using various methods. Of these, a method of acquiring an electrocardiogram (ECG) directly through an attached electrode and a measurement method of using a PPG/blood volume pulse (BVP) sensor on a finger or an earlobe are the most commonly used. The direct method of acquiring an ECG utilizes the principle that the skin senses electrical impulses that occur each time the heart beats. The measurement using a PPG/BVP sensor measures the blood flow and extracts the heartbeat signal from optical electronics by absorbing the infrared light of the cell and blood vessels. The method of measuring and applying bio-signals in this manner has the advantages of simplicity and ease of operation even with the newest smartphones [38].

The heart rate (HR) is the rate of heartbeats, and it is measured as the number of beats of the heart per minute. In other words, HR is the number of peak-to-peak periods in the ECG per minute. One cycle of an ECG consists of QRS waves. The highest peak in one cycle is referred to as the R peak, and the peak-to-peak time interval of each electrocardiogram is called the RR or NN interval or inter-beat interval [39]. A continuous change in the inter-beat interval or RR interval is called heart rate variability (HRV). Because HRV reflects the ability of the heart to adapt to the environment, the analysis of HRV is useful for assessing autonomic imbalances by quantifying the state of the autonomic nervous system, which regulates cardiac activity [40]. Therefore, the activity of the autonomic nervous system activity, that is, sympathetic and parasympathetic nerve activity, can be measured through HRV analysis.

### 2.3. Heart Rate Variability (HRV) Analysis

The analysis of HRV can be divided into analysis in the time domain and analysis in the frequency domain. The simplest method of analyzing HRV is time-domain analysis, in which the time interval of the QRS wave or the instantaneous heart rate at a specific time is analyzed. Analysis in the frequency domain is used to analyze the power distribution of the function to frequency and to quantify the balance of the autonomic nervous system [41,42]. The parameters obtained through the analysis in each domain are called HRV parameters.

In this study, a finger-clip BVP sensor was used to obtain HRV parameters in the time domain and frequency domain. The different variables that can be obtained from the time-domain analysis are the mean RR interval (RRI), standard deviation of normal to normal interval (SDNN), root-mean-square of successive differences (RMSSD) and pNN50(proportion of NN50), but we used the most basic HR and HRV amplitudes. In addition, different variables such as very low frequency (VLF), low frequency (LF), high frequency (HF), VLF/HF ratio, and LF/HF ratio can be extracted through analysis in the frequency domain. The present study used LF, which indicates the activity of the sympathetic nervous system, and HF, which indicates the activity of the parasympathetic nervous system, as parameters, and they were obtained through analysis in the frequency domain. In addition, the LF/HF ratio [43], which can estimate the ratio of sympathetic and parasympathetic activity, was used as a parameter. The detailed meaning of these parameters is described in Table 1.

### 2.4. Emotion Classification Using Machine Learning and Deep Learning

There have been many studies using various parameters of bio signals using sensors. Fujiwara et al. [45] proposed drowsiness detection and validation with HRV analysis and EEG-based signals. Szypulska et al. [46], similar to Fujiwara et al. [45], used HRV analysis to predict fatigue and sleep onset. In addition, research has been proposed to reconstruction PPG signals into ECG signals using MLP. In this study, however, we wanted to classify emotions by the acquired parameters.

Research on classifying human emotions using physiological signals has been performed using various methods and systems. Emotion classification starts with the definition of the emotions to be recognized, the measurement of signals, and the selection of classifiers. In the definition of emotions, the emotions to recognize are selected based on the discrete emotion model or by using a two- or three-dimensional emotion model. Verma et al. [15] and Wen et al. [47] classified human emotions into 13 and 5 emotions, respectively. However, Liu et al. [48] classified emotions by using a two-dimensional emotion model. Additionally, Valenza et al. [49] used two-dimensional emotion models, but they defined emotions in five stages. Theekshana et al. [50] classified discreate emotion as an ensemble model. Thus, even when the study is related to emotion classification, the contents of the research can be changed according to the emotions that are defined.

In the present study, we classified emotions using CNN. Many other studies have classified emotions in other ways. As mentioned in the Introduction, a method of classifying emotions uses conventional statistical analysis techniques and deep learning. Many studies used both—statistical analysis and deep learning—methods or combined them to increase the accuracy of the model. Yin et al. [51] classified emotions into multiple SAEs and DEAP datasets. Their results were compared with the conventional classifier and deep learning, and the performance of the constructed deep-learning model was evaluated. Zheng et al. [19] studied the classification performance of KNN, SVM, and GELM after integrating HMM and DBN. Cho et al. [52] measured human stress using a respiration variability spectrogram (RVS), which was measured with a thermal imaging camera using CNN. In addition to the machine learning and deep learning mentioned above, many studies have been conducted to analyze emotions using fuzzy theory [25,53,54]. Aside from that, deep learning is used in various fields [55].

However, unlike previous studies, the present study proposes an emotion classification into various stages using CNN model. In Section 3, experiments and data pre-processing are introduced. In Section 4, single-signal and multi-signal classification are compared, and a parameter combination procedure is performed between signals. Emotion classification into various stages are determined in Section 5. Section 6 discusses the results. Section 7 concludes the paper.

## 3. Dataset

### 3.1. Experiment

As mentioned in the Introduction, experiments were conducted to classify six emotions and compare the classifiers. It was an open label and single arm experiment performed from November to December 2017 at Seoul Metropolitan Government—Seoul National University (SMG-SNU) Borame Medical Center, Seoul, Korea. Ethical approval was provided by the Institutional Review Board (IRB) at site, and the study adhered to the principles outlined in the Declaration of Helsinki and good clinical practice guidelines. The trial is registered in the Boramae IRB (The Ethics Committee of the SMG-SNU Boramae Medical Center). The experiment was conducted in a controlled environment where a monitor capable of displaying video clips with six distinct emotions and a sensor capable of acquiring physiological signals were installed. To stimulate the participants, video clips expressing happiness, fear, surprise, anger, sadness, and disgust were played for one minute in that order. In order of emotion, the video clips used in the experiment were part of About Time (2013), The Shining (1980), Capricorn One (1977), The Attorney (2013), The Champ (1979), and Pink Flamingos (1972). Snapshots of video clips are not provided in this paper in order not to be restricted by copyright issues.

Before the start of the experiment, participants were sufficiently briefed about the experiment and its side effects. When seated in front of the monitor, the participant put a BVP sensor on the finger and a respiration band on the abdomen. When the experiment began and it was determined that physiological stability was ensured, we measured the physiological signal in the neutral state for 1 min to be used as a different emotion for Control group. Then, participants relaxed for 2 min. After the measurement of the physiological signal of the neutral state and relax, the participant watched the happiness video for 1 min and maintained a relaxed state for 2 min thereafter. When the second relaxed state ended, the participant watched the fear video and the remaining videos in the order indicated in Figure 2. In addition, markers were displayed on the signal at the start of each video and at the start of the neutral state to facilitate data processing.

The software and hardware used in the experiment were products of Mind Media B.V. (Roermond, Netherlands). The BVP signal was from the BVP finger-clip sensor of NeXus-10 MKII, and the sampling rate of the sensor is 32 samples/s. The measurement unit of the BVP sensor is Beats per minute (BPM), and parameters such as Relative Blood Flow and Heart Rate (HR) can be extracted. The measuring range is 40–240 BPM. The wavelength of the BVP sensor IR light is ±940 nm [56]. The respiration signal used the respiration band of the same instrument, and the sampling rate of the sensor is 32 samples/s. BPM parameters can also be extracted from the RSP sensor. The size of sensor housing is 24 mm × 70 mm × 11.6 mm [57]. The used sensors are shown in Figure 3a–c.

The participants wore the BVP sensor as a finger clip. The BVP finger-clip sensor has built-in optical electronics that measures blood flow through the absorption of infrared light in tissue and blood vessels. Every heartbeat causes blood to flow into the arteries and veins. The waveform amplitude of the BVP signal is related to blood flow as well as to the vasodilation and vasoconstriction of blood vessels. The participants wore the respiration band on the chest or abdomen, and the magnetic-field sensor of the band measured the stretching of the band when the participant breathed. The measured band-expansion value is related to the respiration signal. Because this non-invasive and simple measurement method was used to extract physiological signals by using the above-mentioned equipment, the burden on the participants was minimized. In the actual experiment, the sensor mentioned above and shown in Figure 3d was used.

In the BVP signal, various parameters can be obtained to confirm the response of the autonomic nervous system. In this experiment, HRV parameters such as heart rate, HRV amplitude, LF, HF, and LF/HF were extracted. From the RSP signal, the RSP value and RSP rate, which are the RSP parameters, can be extracted. The HRV and RSP parameters in the raw signal were automatically calculated, processed, and stored using Biotrace + software, which did not require manual intervention.

### 3.2. Data Processing

All signals obtained from the experiment can be recorded and confirmed through Biotrace+ software. Based on this software, we selected the data suitable for the study and split them based on the experiment time. From the 53 participants’ signals, 49 signals (30 male and 19 female, 29 ± 10 years old) were used for the study. The remaining four signals were discarded due to signal instability or other reasons. The signal to be analyzed was selected from among the HRV parameters (heart rate, HRV amplitude, LF, HF, and LF/HF) and RSP parameters (RSP value and RSP rate), which are stored as time-series data. Subsequently, we split the data according to the markers, as recorded in the experiment, and extracted the data corresponding to the six emotions for 1 min. Data processing was completed by labeling each of the six extracted signals with their corresponding emotion. The overall procedure of data processing is represented in Figure 4. The data obtained in this manner were applied to the pre-processed data to be used in the classification model presented in Section 4.

The number and type of data obtained through data preprocessing procedures are summarized in Table 2.

Although the number of all data is specified here, the data used in the actual analysis may vary slightly depending on the individual characteristics of the experiment participants. The data used for analysis depend on the combination of features. The combination of features can also be found in Section 4. In addition, information about the number of features can be found in Table 3.

## 4. Method

The pre-processed data obtained from the data-processing step were applied to the classification of emotions. Although emotions can be classified in various ways, in this study, we compared results of single-signal and multi-signal classification and classified emotions into various stages. For this study, we constructed a CNN model and applied the pre-processed data. The overall procedure of comparing single-signal and multi-signal from data acquisition from each sensor is shown in Figure 5.

The obtained HRV- (HR, HRV Amplitude, LF, HF, LF/HF Ratio; HRV) and RSP-parameter (RSP Value, RSP Rate; RSP) data were applied to the CNN as parameters to classify the emotions. Each CNN model used emotion- (happiness, fear, surprise, anger, sadness, and disgust) and neutral-labeled data as input data to classify a given emotion.

### 4.1. Parameter Combination

The phased combination of each parameter was proceeded to find the optimal classification model. The combination of all physiological signal parameters obtained through experiments is depicted in Figure 6.

Combination steps (1) and (2) correspond to single-signal classification (RSP parameter or HRV parameter). Combination steps (3) and (4) correspond to multi-signal classification (combination of RSP and each domain of HRV parameters). Combination step (5) corresponds to multi-signal classification (RSP and HRV parameters).

Particularly, Combination step (3) was applied to the classification model through the fusion of the RSP parameters and the frequency-domain HRV parameters. Similarly, Combination step (4) was applied through the fusion of the RSP parameters and the time-domain HRV parameters. These step-by-step combinations of parameters were intended to verify the optimal emotional classification model.

### 4.2. Convolution Neural Network Model

CNN is generally used for the classification and prediction of images. However, by modifying the structure of CNN, we used a method of row-by-row reading of input data to classify physiological signals. CNN used Python’s frameworks, Keras, and Tensorflow. CNN utilized 70% of all data as a training set and the other 30% as a test set. Figure 6 shows how the CNN model read the seven-parameter input data for emotions classification. Each row was used as input data and each column was reduced as it went through the convolutional layer. In particular, Figure 6 illustrates how the shape of the input data changes from (7 1 1) to (6 1 1). If the input data were changed to 2 or 5, the columns in Figure 7 would be changed, respectively.

However, when the Combination steps (1)–(5) were classified using the CNN, the input data varied according to the number of input parameters. That is, when classifying the RSP: Combination step (1), the model used the input data shape (2 1 1), while Combination step (2) with HRV used (5 1 1). Thus, for RSP and HRV (Combination step (5)), (7 1 1) was used. Overall, the model was constructed to have similar tendencies in large frameworks. The input data shapes are listed in Table 3, and the constructed CNN model is shown in Figure 8.

The tendency of the overall CNN model constructed in this study was designed by reducing the size of the shape by one (*n*, *n-1*, *n-2*, *n-3*, etc.). Therefore, in Model 1, the input shape was (2 1 1) because two input parameters were applied, and the shape decreased as learning progressed. Models 2–4 were constructed to have the same tendency. Furthermore, in Models 2–4, we reduced the number of convolution filters from 50 to 25 and then increased it to 50. In all models, batch normalization and activation functions were applied between two convolution layers. Finally, maxpooling proceeded. We also initialized weights with the “He kernel initializer” function in all convolution layers. Finally, a He kernel initializer was used in the ReLu and Softmax functions along with Dense, and the dropout rate was set to 0.5.

The performance of the CNN model depends on the hyperparameters as well as the shape of the constructed model. To classify the six emotions in the constructed model, different learning rates and batch sizes for each model were applied to the CNN model constructed using the hyperparameters listed in Table 4.

The contents described in Section 3 and Section 4 are procedures of acquiring raw data, pre-processing data, selecting parameters, combining signals, and building a classification model. This is the most important part of this study and the essential part of the process of finding the most novel and optimized emotion classification models. Figure 9 shows the pipeline of the overall process.

## 5. Results

Before mentioning the accuracy of the classification results, we present the method used for calculating accuracy in this study. There are two major cases of classification using deep learning. One is condition positive and the other is condition negative. In general, positive means identified and negative means rejected. Each condition positive and negative is divided into true and false. True means correct and false means incorrect. Therefore, the results of classification using deep learning are classified into four types: true positive (TP), false positive (FP), true negative (TN), and false negative (FN). In these four cases, the formula for accuracy is shown in Equation (1).
(1)$$\mathrm{Accuracy}=\frac{TP+TN}{TP+FP+TN+FN}.$$

We classified the parameters into six emotions based on the CNN models described in the preceding subsection and evaluated the single-signal and multi-signal classification performance. The classification results are summarized in Table 5 and Table 6.

The classification accuracy was 63.18% and 77.42% when emotions were classified using the single-signal with CNN Models 1 and 2, respectively. This result suggests that both classifiers can classify emotions with proper performance when using the HRV parameters and show better results compared to classification using the RSP parameters alone. The results of the classification were identified, but the results contained some meanings that were insufficient to be used as a model for general emotion classification. Therefore, instead of classifying emotions using a single-signal, we verified the classification results of emotions using multi-signal. Table 6 shows the results of classifying multi-signal.

The classification accuracy was 78.31% and 76.02% when parameters which combined RSP and some HRV were classified using the multi-signal with CNN Models 2 and 3, respectively. This result suggests that both classifiers can classify emotions with proper performance when using the parameters which combine RSP and some HRV parameters. Unfortunately, while CNN Models 2 and 3 were used to classify emotions properly, similar to some of the results in Table 5, it was rather insufficient to actually classify general emotions. Finally, the accuracy was 94.02% when emotions were classified using all the multi-signal parameters with CNN Model 4. This result suggests that CNN Model 4 can classify emotions with superior performance when using the RSP and HRV parameters.

These five cases reveal that multi-signal classification is better than single-signal classification. However, not all multi-signal models have outstanding results, and multi-signal consisting of only some domains of HRV did not yield high classification results. Therefore, this study confirms that the accuracy of multi-signal emotion classification is higher than that of single-signal emotion classification. When classifying emotions using multi-signal, all domains of the HRV parameter must be used.

## 6. Discussion

Various physiological signals can be detected in the human body, and each physiological signal can describe various states of the body. In addition, since physiological signals cannot be changed at will, it is a good indicator for analyzing a person’s emotional state. Much research has been conducted on emotional analysis [9,20,58,59,60], and this research was based on a two-dimensional emotion model or a few discrete emotion models. In addition, emotion classification was performed with respiration signals and ECG signals [30,31,61], and multi-signal classification using signals such as EEG, GSR, EMG, and SKT, which are less relevant to respiration, has been used [10,62]. However, the dimension emotion model actually has a difference in feeling between the six basic emotions in daily life. Therefore, in this study, we studied emotion classification based on the six emotions with a focus on optimal emotion classification model, unlike previous studies that classified emotions simply.

The optimal emotion classification model can reduce unnecessary procedures in analysis and experimentation. Therefore, we compared the accuracy of multi-signal and single-signal classification, performed classification using combined physiological signals into various stage, and then judged the accuracy of emotion classification using some parts of cardiac and respiratory signals.

A total of 53 subjects participated in the experiment. They watched video clips corresponding to the six basic emotions while wearing a finger-clip BVP sensor and chest-band respiration band. The BVP and RSP signals of the subjects were measured while they watched the video clips. Of the raw signals, 49 signals were used in the study, removing inappropriate data. The signal selecting and splitting proceeded according to the research purpose. Subsequently, labeling and data pre-processing were performed.

The labeled data obtained in the pre-processing step were applied to the classifier. RSP (single signal) and HRV (single signal) were applied to CNN to identify single-signal classification performance. For this purpose, two CNN models (CNN Models 1 and 2) were created for each parameter. Through the results, when single-signal was used, it was confirmed that proper emotion classification performance can be obtained. However, it cannot be used for an independent emotion classification model because of the low performance.

To solve this problem, we decided to classify emotions through the combination of different domains of HRV or all of HRV and RSP. Combining different types of signals can improve classification performance. The signals were combined in three ways: the combination between the RSP signal and frequency-domain HRV signal; the combination between the RSP signal and time-domain HRV signal; and the combination between the RSP signal and HRV signal. These were named Combination steps (3)–(5), respectively.

The combined signal was classified with CNN in the same manner as in the previous cases. As a result of classification using Combination steps (3) and (4) signals with CNN, the accuracy was confirmed to be similar for the two classifiers. In addition, Combination step (5) results show an accuracy greater than 94%; thus, it was confirmed that Combination step (5) yielded higher performance with all classifiers. Multi-signal classification showed the better performance than single-signal classification. Therefore, it can be said that the use of many types of input signals can enhance the emotion classification performance. However, when using multi-signal, all domains of HRV must be applied as input data.

However, in the combination of RSP and some of HRV signal, the results of emotion classification are similar. In other words, the accuracies of Combination steps (3) and (4) are similar. We identified the dominance of the entire HRV parameters to verify why both results are similar. Therefore, we performed principal component analysis (PCA) for all the HRV parameters. Table 7 shows the PCA results for all the HRV parameters. PC1 and PC2 can explain 65.3% of the total data. The cumulative proportions of the PCA are also listed in Table 7.

Based on the results shown in Table 7, a correlation circle was drawn, as shown in Figure 10, to confirm the dominance of the entire parameters. The length of the arrow corresponding to each variable indicates the contribution of that variable and the relative magnitude of its variance. In other words, the contribution of each parameter to the PC is shown in Figure 10.

As shown in Figure 10, the most significant influence on PC1 is the HRV frequency domain parameter; similarly, the HRV time domain parameter is the most significant influence on PC2. When looking at the length of the arrow, the length of the arrow between the two domains is not much different. Based on this, for each domain of the HRV parameter, it can be verified that there is no dominant domain on either side. Therefore, we can see why the classification results of Combination steps (3) and (4) are similar.

Based on the above studies, we can determine that the optimal model for classifying emotions using multi-signal is CNN Model 4, and it is difficult to classify general emotions with single-signal or some combination of data. Therefore, when classifying emotions, both RSP and HRV parameters should be used.

By modifying the structure of the CNN model, we developed a general signal classifier. The data were acquired through six emotion-based video viewing experiments and preprocessed. Four different CNN models were developed for preprocessed data. Based on this, the classification accuracy of single- and multi-signal was compared. In addition, emotions were classified at various stages by signal combinations, and the dominance of HRV parameters was verified to confirm that it is difficult to classify emotions with multi-signal combined with several domains. Finally, an optimal model for classifying emotions with high accuracy is presented.

## 7. Conclusions

The response of the human autonomic nervous system is a good indicator of emotions because it cannot be manipulated at will. In this study, we studied the optimal emotion classification models. Unlike previous studies using many physiological signals for emotion classification, the present study attempted to obtain an optimal emotion classification model. Respiration and cardiac signals corresponding to six basic emotions were extracted through experiments; the signals were combined variously to find the optimal classification model; and several CNN models were built to classify emotions. PCA was also used to determine the cause of similar emotional classification results.

Although multi-signal emotion classification shows very good results, it takes a relatively long time because of the many parameters and data used, and the calculation requires high computing power. Therefore, further research should be conducted to improve the performance of emotion classification with fewer parameters and advanced CNN models.

Finally, signals were measured for one minute in the experiment. If the constructed model were commercialized, this may be a rather long time. Since classification results may vary depending on the measurement time of the signal, further research will be required to shorten the time of signal extraction and improve the model accuracy. Although there are limitations in the present research, it is possible to develop a more efficient emotion estimation technology from the results of this study.

## Figures and Tables

**Figure 1 sensors-20-00866-f001:**
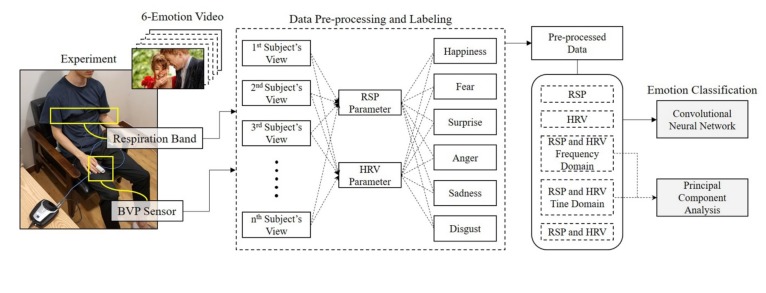
Overall flow of the study.

**Figure 2 sensors-20-00866-f002:**
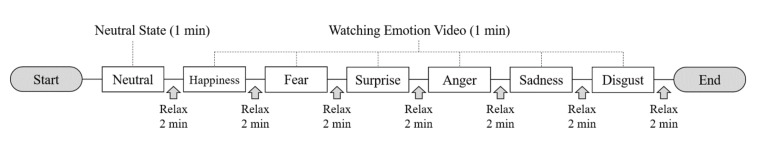
Experimental procedure.

**Figure 3 sensors-20-00866-f003:**
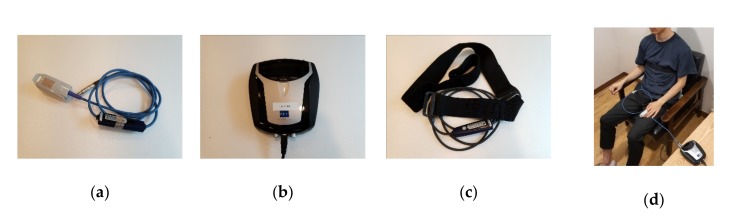
Experimental equipment: (**a**) finger-Clip BVP Sensor; (**b**) NeXus-10 MKII; (**c**) chest-Band Respiration Sensor; and (**d**) signal measurement.

**Figure 4 sensors-20-00866-f004:**
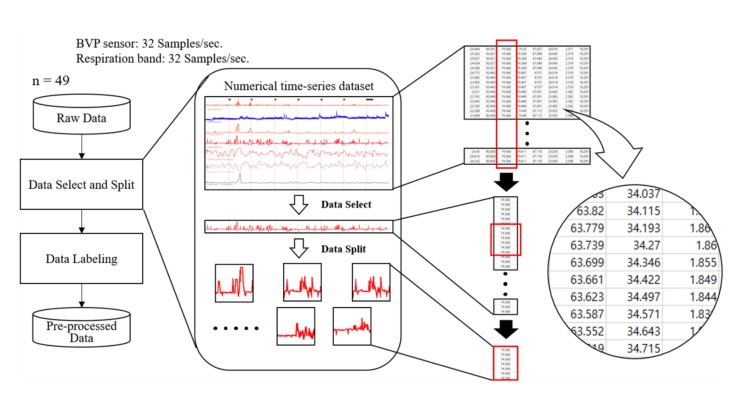
Data pre-processing procedure.

**Figure 5 sensors-20-00866-f005:**
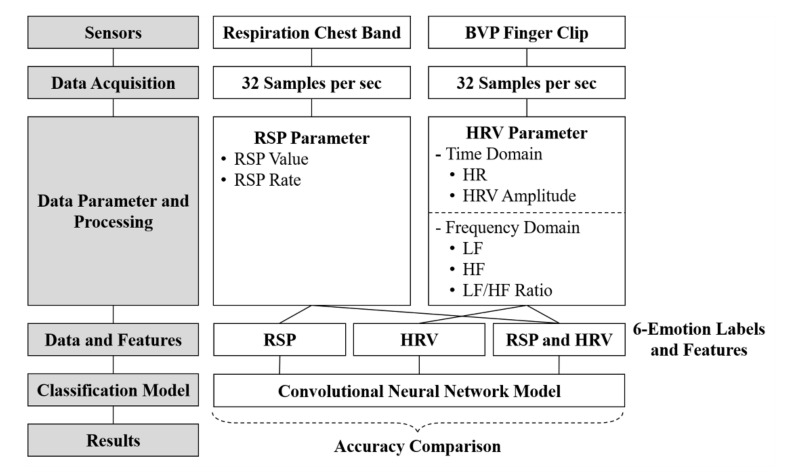
Procedure for comparing and selecting the classification model.

**Figure 6 sensors-20-00866-f006:**
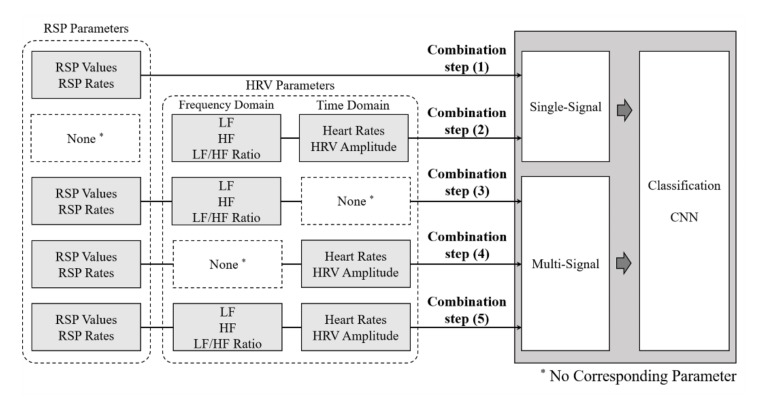
Parameter combination method of physiological signals.

**Figure 7 sensors-20-00866-f007:**
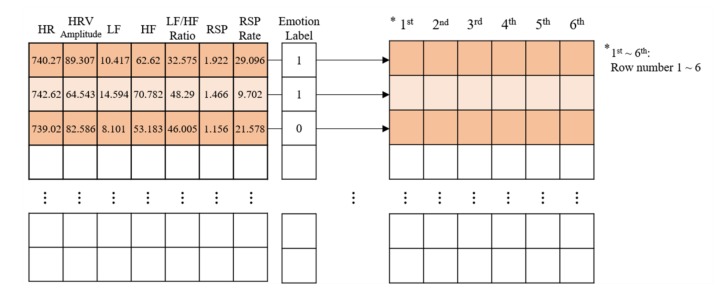
Method of processing input data of the CNN model constructed in this study.

**Figure 8 sensors-20-00866-f008:**
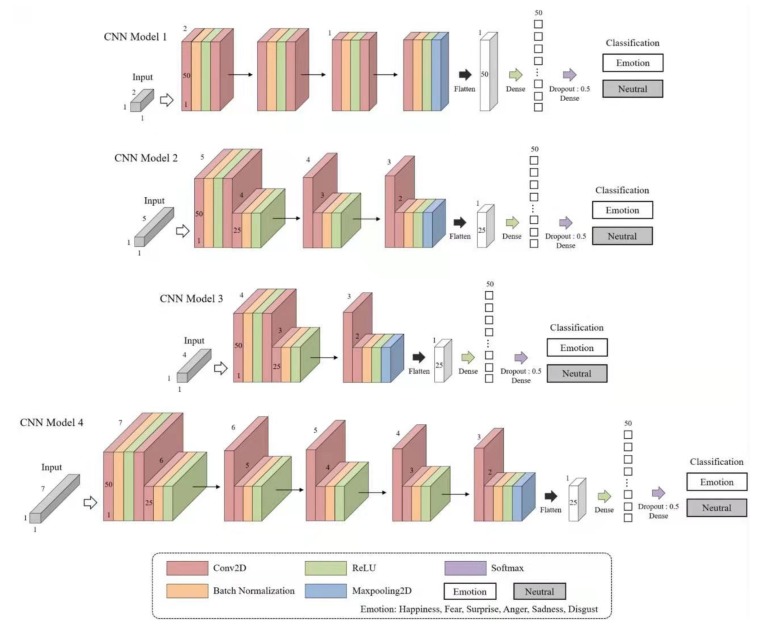
CNN Models 1–4, which were constructed differently according to input data.

**Figure 9 sensors-20-00866-f009:**
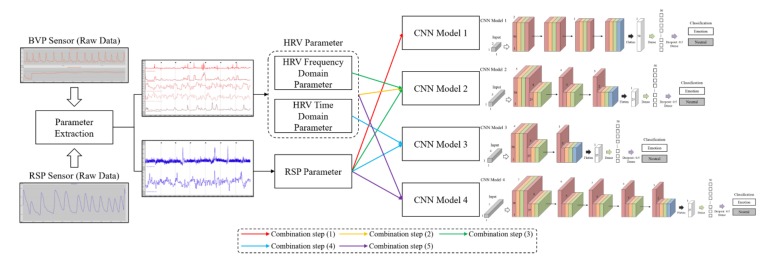
The pipeline of the overall process.

**Figure 10 sensors-20-00866-f010:**
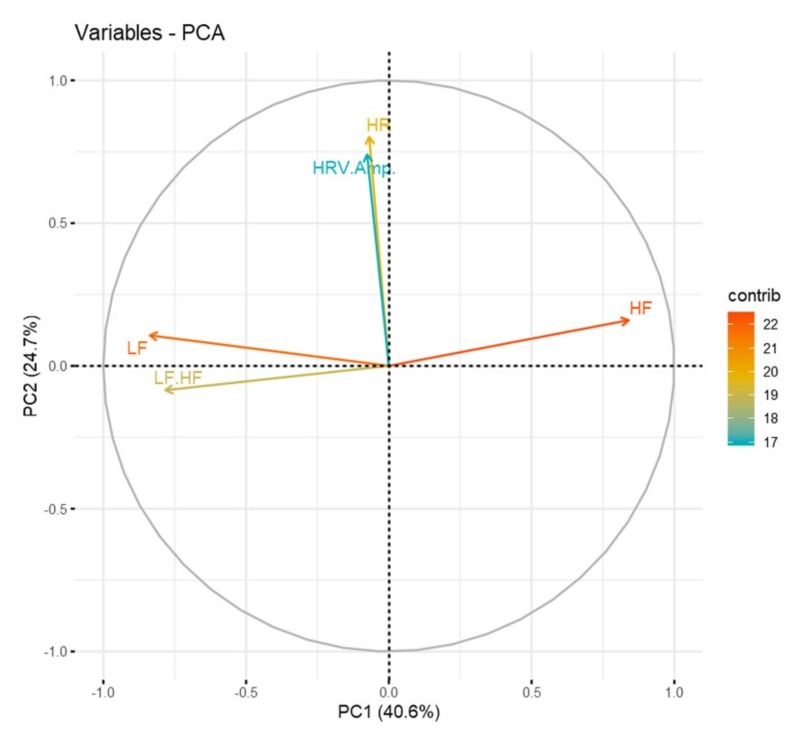
Comparison of parameters based on correlation circle.

**Table 1 sensors-20-00866-t001:** Frequency-domain Heart Rate Variability (HRV) parameters [44].

Features	Description
LF	Power in low frequency range (0.04–0.15 Hz)
HF	Power in high frequency range (0.15–0.4 Hz)
LF/HF Ratio	LF/HF

**Table 2 sensors-20-00866-t002:** Total amount of data used in the study (Units: Data).

Time (s)	Six Emotions and Neutral State	Sampling Rate	Number of Participants	Number of Input Parameters	Total Amount
60	7	32	49	2	1,317,120
60	7	32	49	5	3,292,800
60	7	32	49	7	4,609,920
60	7	32	49	4	2,634,240
-					11,854,080

**Table 3 sensors-20-00866-t003:** Variation of input shape depending on input signal.

Input Parameter	Number of Input Parameters	Model No.	Input Shape
RSP: Combination step (1)	2	CNN Model 1	(2 1 1)
HRV: Combination step (2)	5	CNN Model 2	(5 1 1)
RSP and HRV Frequency Domain: Combination step (3)	5	CNN Model 2	(5 1 1)
RSP and HRV TimeDomain: Combination step (4)	4	CNN Model 3	(4 1 1)
RSP and HRV:Combination step (5)	7	CNN Model 4	(7 1 1)

**Table 4 sensors-20-00866-t004:** Hyper-parameters applied to CNN Models 1–4.

Input Parameter	RSP	HRV	RSP and HRV Frequency Domain	RSP and HRV Time Domain	RSP and HRV
**Applied Model**	**CNN Model 1**	**CNN Model 2**	**CNN Model 2**	**CNN Model 3**	**CNN Model 4**
Batch Size	64	16	32	32	16
Learning Rate	0.00001	0.0001	0.0001	0.0001	0.0001
Optimizer	Adam Optimizer				
Cost Function	Cross Entropy				

**Table 5 sensors-20-00866-t005:** Accuracy of single-signal for six emotions classification.

-	Happiness	Fear	Surprise	Anger	Sadness	Disgust	Average
RSP (CNN Model 1)	64.24	61.61	60.38	66.7	64.08	62.04	63.18
HRV (CNN Model 2)	75.3	75.34	78.28	78.21	77.91	79.49	77.42

**Table 6 sensors-20-00866-t006:** Accuracy of multi-signal for six emotions classification.

-	Happiness	Fear	Surprise	Anger	Sadness	Disgust	Average
RSP and HRV Frequency Domain (CNN Model 2)	79.77	77.81	76.49	80.86	78.52	76.39	78.31
RSP and HRV Time Domain (CNN Model 3)	75.83	75.75	76.31	79.74	72.9	75.59	76.02
RSP and HRV (CNN Model 4)	93.69	95.83	93.08	95.57	94.11	91.82	94.02

**Table 7 sensors-20-00866-t007:** Importance of principal components.

-	PC1	PC2	PC3	PC4	PC5
Standard Deviation	1.4250	1.1121	0.9004	0.7414	0.6101
Proportion of Variance	0.4061	0.2474	0.1621	0.1099	0.07445
Cumulative Proportion	0.4061	0.6535	0.8165	0.9255	1.0000

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
