# Peer review of "The Design of CNN Architectures for Optimal Six Basic Emotion Classification Using Multiple Physiological Signals"

_sensors, 2020, doi:10.3390/s20030866_

Round 1

Reviewer 1 Report

This work lacks of a comparison experiment, that is, a performance comparison between this work and previous relevant studies to show the advantages and disadvantages of this method. The comparison study should be conducted on the same dataset the authors established.

The innovation of this paper is not clear. The authors must make it clear that where is the main innovation of this paper? Experimental paradigm, multimodal data set, or CNN network?

As for the multimodal physiological signals, are the signals of HRV and BSP sufficient? The authors need to state the reason why these two signals were chosen. Why not add other signal acquisition in the experiment?  

Reviewer 2 Report

The revised paper seems good. Only few English checks are requested.

Reviewer 3 Report

The manuscript is well written and centered on an interesting topic. Organization of the paper is good and the proposed method is quite novel. The manuscript, however, does not link well with relevant literature on multimodal sentiment analysis, e.g., see sentic blending and works on real-time multimodal sentiment analysis. Also, check recent literature on fuzzy commonsense reasoning for multimodal sentiment analysis.

Reviewer 4 Report

The authors bring an interesting approach to the problems of emotion classification. However, from the perspective of the review, I have several comments that must be incorporated in this research paper.

Reminder 1: In the section Introduction, the authors state: „Voice-based emotion recognition identifies instantaneous emotion with high accuracy. However, it has the disadvantage of recognizing emotions only over short time periods“. The various researches on voice processing and subsequent classification of emotions (Magdin, Martin & Sulka, Timotej & Tomanová, J. & Vozar, Martin. (2019). Voice Analysis Using PRAAT Software and Classification of User Emotional State. International Journal of Interactive Multimedia and Artificial Intelligence. DOI: 10.9781/ijimai.2019.03.004. ) indicate that the overall success of the classification is strongly dependent mainly on databases that are designed for neural network training. Currently, there are only a small number of these databases that are freely accessible, respectively after registration (Ravdess dataset, EMO-DB, Save and otrhers). However, these databases contain an insufficient amount of input data to the neural network. Therefore the input data must be modified eg. frequency change, formants, and the like. Also, the input sample is strongly dependent on the language in which the audio sample is made. For example, in Spanish or English, the voice output from neural net may be classified as neutral, in the case of Chinese; the sample may be classified as surprise or anger. Therefore, it is important to always combine multiple sensors for detection in order to identify and then classify the right emotional states.

Reminder 2: Measurement method using finger-clip blood volume pulse (BVP) sensor, HRV and RSP sensors is an invasive measurement method. The problem with this type of measurement is that the respondent is aware of the measurement and can be controlled (self-control) and the results may not be accurate. The authors did not give the respondents psychological questionnaires they would investigate this situation.

Reminder 3: The authors used the following parts of film clips: About Time (2013), The Shining (1980), The Capricorn One (1977), The Attorney (2013), The Champ (1979) and Pink Flamingos (1972). Why didn't the authors use any of the standardized video databases that include video clips classification by emotional state, valence, excitement and other important factors?

Reminder 4: Why the authors used just 2 minutes for each respondent. Did every respondent calm down in these 2 minutes?

Reminder 5: Did authors the respondents ask how they felt when watching a particular video clip? Like did the authors compare (the measured) data?

Reminder 6: In this research paper authors does not that was the aim in a specific movie trailer. The authors did not determine the expected emotions from the video clips, their valence and the dimension of arousal that respondents could experience.

Reminder 7: However, since the data are not compared (see observation 6), they cannot be considered relevant. The data must always be compared with the reference dataset.

Round 2

Reviewer 1 Report

Line 52-53 The sentence should be rewritten for a clear understanding.

Line 214-224 The descriptions of completed experiment should be in the past tense.

Line 274 The unit of these numbers should be listed in Table 2.  

Line 291-292 please double check the HRV-Parameter and RSP--Parameter descriptions.

Table 5-6 The authors should declare how to calculate the accuracy, and how many rounds (code running) lead to the accuracy (is the accuracy the maximum or the average value?).

Reviewer 4 Report

All questions was correct answered.

Author Response

This manuscript is a resubmission of an earlier submission. The following is a list of the peer review reports and author responses from that submission.

Round 1

Reviewer 1 Report

The manuscript presented a convolution neural network model for efficient classification of six-basic emotions using multimodal physiological signals. For this paper, in order to effectively recognize emotions from respiration and electrocardiography, two methods (SVM and CNN) are used for comparison. Besides, the method of multimodal fusion is used to obtain high classification accuracy with a small amount of data in this manuscript. Furthermore, this paper also studies the influence of parameter precision on the recognition model, and proposes a method to analyze the input data through statistical analysis among models with similar results. The paper can reflect the workload, which is worth affirming. The results proved the effectiveness of the proposed method to some extent. However, the manuscript has several critical limitations:

Weak motivation:As far as I know, four physiological signals (electrocardiogram (ECG), skin temperature (SKT), skin conductance (SC) and respiration) have been used in the article “Emotional Recognition using Physiological Signals” for emotional recognition as early as 2006.So,why did the authors use only two physiological signals, heartbeat and respiration? As you said in this article introduction,” numerous studies have shown that physiological signals and human emotional states are closely related”. Honestly, the current identification of human emotions is mainly using the EEG and it has been proves that the EEG signal contains more emotional information than any other physiological signals. Why do not choose it? The title of the article does not match the content: The title of your article is “ The Convolution Neural Network Model for Efficient Classification of Six-basic Emotions using Multimodal Physiological Signals”. I have the following questions about the title:

(1) A new method to identify physiological signals is proposed in this paper. However, the recognition effect of the new method in this paper is not as good as that of SVM. In that case, why is CNN proposed as a new method in this manuscript?

(2)The title in the manuscript mentioned that emotional recognition was conducted by fused physiological signals. The statistical analysis results show that a single HRV signal on SVM can effectively identify emotions and compare fusion signals. Is the title of the article inconsistent with the content of your research?

(3) Besides, the recognition results of single HRV and fusion (a) in CNN were not different. In order to get a good recognition model, a large amount of data needs to train in the CNN compared with less data required by SVM. Whether the acquired data can be well verified in the deep learning model?                                                                                                                              

In addition, some minor concerns are list below:

Line 37, Page 1. ’such as and depression’. Dose the first sentence omit anything after “such as” ? Before section 5.2, was the input data processed by model 4 fused? How to arrange the order of the data which input into PCA? What does the element ( such as PC1) mean? The convolution kernel in a convolutional neural network that recognizes physiological signals is usually smaller than the input data. Why does the convolutional neural network in this paper design the size of the convolution kernel large than the input data?

In my opinion, the current manuscript is below the acceptance bar due to limited novelty. If the authors can address the problems raised before, the manuscript can be reconsidered again.

Reviewer 2 Report

The emotion stimulation materials are not standardized. SAM test should be conducted to the materials by at least 20 subjects for validation of the emotion tag (VAD value or emotion type) of the stimulation video clips. There are problems in the design of experimental paradigm, which must affect the data and analysis.

1) Neutral is also an emotion state, therefore a relax state should be applied between neutral and other emotion stimulations.

2) The seven emotion states (neutral, happy, fear, surprise, anger, sad, disgust) must be randomly assigned to eliminate the influence of emotional order on the subjects.

The abstract announced that 114 participates were involved, why only 20 subjects data were used? The data amount is too small. Page 8, CNN Model 1: compared with the other two models, maxpooling layer seems useless in the network structure. what does it mean by getting maximum from only one number? Page 8, 11, CNN Models: two categories E and N are shown in the figure, which are not corresponding to the 6 emotions classification task mentioned in the paper. It is necessary to make clear that how many categories are involved in the task, 2 or 6 or 7(with neutral included) ? In figure 5, different feature orders might affect the networks’ performance, since convolutional operations are highly relevant to the orders. But this problem is not considered in the CNN models. Try to discuss the physical or physiological or mathematical meanings of figure 5, where the authors put different physiological data into a matrix. Line 37: such as ____ and depression? Line 60: deep running or deep learning? The references should be updated, it is better to include new references in the year 2018 and 2019.

Reviewer 3 Report

The authors propose a novel emotion recognition method using multimodal 14 physiological signal parameters acquired by bio-signal sensors for improving the efficiency of 15 classifying individual emotional responses. The method is based on the study and post-processing of the RSP and HRV physiological indicators determined on a dataset of 114 patients. Then using a convolutional deep learning framework and SVM processing, they are able to classify the emotions of the analyzed subjects.

The methods seems promising but the authors must pay attention to some parts.
The references of known-art are poor in the part concerning deep learning methods for the analysis and stabilization of physiological signals. Similar considerations with regard to the part of sensors used for signal acquisition.

The authors arte suggested to add more details about used hardware, boards, sensors, signals and methods used to sample and stabilize the acquired data. After that, It is suggested to the authors to extend this part better, also inserting a careful scientific comparison with the following papers:

Fujiwara, K .; Abe, E .; Kamata, K .; Nakayama, C .; Suzuki, Y .; Yamakawa, T .; Hiraoka, T .; Kano, M .; Sumi, Y .; Masuda, F .; et al. Heart rate variability-based driver drowsiness detection and its validation with EEG. IEEE Trans. Biomed. Eng. 2019, 66, 1769–1778

Vinciguerra, V .; Amber, E .; Maddiona, L .; Romeo, M .; Mazzillo, M . ; et al. PPG / ECG multisite combo system based on SiPM technology. Lect. Notes Electr. Eng. 2019, 539, 105–109.

Mazzillo, M .; Maddiona, L .; Rundo, F .; Sciuto, A .; Libertino, S .; Lombardo, S .; Phallic, G. Characterization of SiPMs with NIR long-pass interferential and plastic filters. IEEE Photonics J. 2018, 10, doi: 10.1109 / JPHOT.2018.2834738.

Szypulska, M .; Piotrowski, Z. Prediction of fatigue and sleep onset using HRV analysis. In Proceedings of the IEEE the 19th International Conference Mixed Design of Integrated Circuits and Systems — MIXDES, Warsaw, Poland, 24–26 May 2012.

Rundo, F .; Petralia, S .; Fallica, G .; Conoci, S. A nonlinear pattern recognition pipeline for PPG / ECG medical assessments. Lect. Notes Electr. Eng. 2019, 539, 473–480.

The authors should improve reported figures as a blurring effect is evident in many of them.
From figure 6, in the pipeline of used CNNs it is evident the use of a dropout of 0.5 to avoid such overfitting issue of the system. Have the authors tried using different Dropouts? If so, with what results ? Add details.

In section 4.2, the performances of the three approaches based on the use of CNNs networks with different architecture are reported, demonstrating that the model 3 is better in terms of accuracy in the classification (if the SVM approach is excluded). However, the CNNs model 3 seems to be the most complex from the architectural point of view, therefore it is necessary to add a computational cost analysis of the three models.

Moreover, it is difficult to me to understand that SVM performs better than CNNs but I can suppose that it is related to the used/tested CNNs which are not very complex. Have the authors tried with more efficient CNNs model such as ResNET or like that ?

Add details.

Reviewer 4 Report

This study presents a comparison of single-signal and multi-signal machine learning and deep learning methods using physiological signals for emotion classification. The evaluation was performed on a dataset collected from 20 subjects (114 participants but uses only 20 for the paper) using respiration and heart rate signals. 

One of the positive things regarding this paper is the experimental setup
with clear protocol for eliciting emotions, 3 sensors and 114 participants. However, the data is not available to the reviewers, therefore, I can’t confirm the claim of the authors about the data. Also, the authors chose only 20 subjects for their experiments which I think is little (only 20 minutes per emotion). 

One major comment is that the goals of the study are not clearly formulated. The authors claim that they propose an "efficient emotion classification method beyond simple emotion classification", but it is not clear what they mean with this. How is it efficient? Why are other studies not efficient? Besides simple classification, what else is analyzed and how is it evaluated? Although the evaluation encompasses an analysis of factors for the classification, the method still only classifies emotions with single or multiple signals so it is not clear what "beyond classification" is proposed. As it is, the novelty is very low.

A second major comment is the structure of the paper. The writing is streamed, and in many places claims are repeated without further explanation. For example, in line 149 and 168, it is stated that this study uses CNN but there is no explanation to why it is used, why it should be used, what are the advantages, what is proposed in this paper besides the use of CNN, etc. It is also not clear the connection with previous sentences. The related work section needs to be structured and the analysis needs to be improved. Currently, it summarizes other works without stating differences with this study. In terms of structure, it begins by saying how respiration signal is used and only until line 161 it says that the respiration signal is a physiological signal. So it is a little bit hard to read and understand. The introduction also needs to be improved to clearly define the problem, goal and novelty of the study. The text is not cohesive. 

It is not clear how the results of the study may be applied to develop medical treatment programs. 

Other points: 

Figure 5 is not clear. Do the colors have some meaning like RGB in an image? If CNN expects and image (2D matrix with one label), how did you construct one 2D sample from one row? What label was assigned? Please further clarify this point. 

In line 263, it is not clear what you been by applying the RSP and HRV data as parameters of the SVM and CNN. Do you mean as input data? 

Some misspellings:

-line 328 multimod e l 

- line 37 such as [missing] AND depression

-line 185 experiment

Round 2

Reviewer 1 Report

I think the main concern towards this artcile is the innovation. The revised title is 'the design of convolutional neural network' . I think that the authors want to describe that this is the innovation; however, this cannot bring benefits at least in your experiments even it cannot beat SVM.

The data preparation and the feature extraction is confusing. 

Reviewer 2 Report

Subject number of this paper is confusing. The authors stated that they recruited 112 subjects in the abstract, while in the comments reply and manuscript the statement sometimes changed to 32 subjects and sometimes 20 subjects. This statement is not consistent and the writing is really not careful. The authors replied in their comments that there was a rest break between neutral and happy, however in Figure 1 of the revised paper, it can’t be found. The authors used sampling rate (points/second) to calculate the data amount, which was not appropriate. Usually we used 5-second long data for feature extraction, or 1-second long data to enlarge the data amount. However we can’t use only one data point for calculating For example you can not calculate HR or HRV by only one data point. The estimation of data amount in table 2 is incorrect, which should be carefully checked and reorganized. Fixed sequence emotion stimulation experiment paradigms might bring interference in the result. In the emotion recognition experiment, it should be considered carefully. The paper didn’t present a strong evidence that the presented CNN model was better in applications. At least in this dataset, SVM was more recommended. Moreover, the training of CNN needs more data, where 20 subjects data might not be adequate.

Reviewer 3 Report

The authors have not adequately applied the proposed suggestions given regarding both the papers mentioned and the comparison with the prior art on the methods that make use of the Deep Learning pipeline.
Authors are advised to note the given suggestions by adding more detailed descriptions of the hardware setup used with respect to the known art in this area (see suggested papers).

Furthermore, better specify the reasons why SVM would seem to perform better than the proposed CNNs model.

Reviewer 4 Report

The authors have addressed my concerns and the paper has improved much. Although the data is small, the results are interesting for the research community and the recommendations provided are grounded in these results.